# ShreddingNet: Coarse-to-Fine Restoration for Multi-Source Shredded Manuscripts

## Abstract

As an important research task of human cultural heritage, the restoration of artworks and calligraphy is of great significance. Seldom existing works have taken the multi-source (*i.e.*, fragments are not ensured to be from the same piece of artworks) fragment-oriented restoration task into account. In this paper, we introduce a restoration algorithm for shredded artworks based on a coarse-to-fine two-stage pipeline. This is an algorithm that can handle the multi-source shredded artworks restoration problem without any restrictive conditions or strong assumptions, and it admits a linear time complexity and robustness to stains, mold, and contour defects. In the proposed coarse matching stage, the algorithm compares the features of each fragment, generating candidate matching pairs. Although a significant number of erroneous matching pairs persist in the candidate set, erroneous matches between fragments from different source images are often rare, enabling high-accuracy clustering of fragments belonging to the same image. In the introduced fine-grained matching stage, the algorithm filters out erroneous matching pairs from the candidate set, producing more precise final matching pairs for global assembly. Experiments conducted on more than 4,000 images from two datasets demonstrate the average reconstruction F1-score achieves 98.37%, which is 5.72% higher than the current state-of-the-art method, confirming the method's effectiveness and robustness. Source code is available in the supplementary material.

## 1 Introduction

Artworks and calligraphy are an invaluable cultural heritage of humanity, possessing a profound historical, cultural, and artistic value. Traditionally, many precious artworks and calligraphy have suffered from fragmentation due to prolonged time or human-induced harm. The rapid development of vision technologies has provided new technical approaches for the digital restoration of manuscripts (Funkhouser et al., 2011b; Sommerschield et al., 2023) (here after this paper will use the term "manuscript" to refer to the calligraphic works and painting artworks etc. of interest in this task).

In recent years, some research progress has been made in the task of restoration from ancient manuscripts and cultural heritage, such as oracle bone pieces (Zhang et al., 2022a; 2021; 2022b), Dunhuang manuscripts (Yu et al., 2022; Zhang et al., 2025), and ancient papyrus (Abitbol et al., 2021). Few studies address the multi-source(*i.e.*, fragments are not ensured to be from the same manuscript) manuscript restoration tasks discussed in this paper. JigsawNet (Le & Li, 2019) introduced convolutional neural networks to screen fragment pairs in single-source tasks, where all fragments derive from one image, but it and its related works (Bridger et al., 2020; Son et al., 2019; Pomeranz et al., 2011) struggle with multi-source tasks involving fragments from multiple images. LLMCO4MR (Zhang et al., 2025) employs a graph neural network with an optimal transport layer to remove a fixed number of outlier fragments in multi-source scenarios, though its assumption of a known outlier count is often impractical. PairingNet (Zhou et al., 2025) aims at identifying the adjacent fragments of a given fragment within a multi-source fragment dataset. This method focuses on identifying adjacent fragments for a given fragment and is not designed for the task of restoration of the entire image. Its identification results exhibit high recall but low precision, limiting its effectiveness for complete restoration. Many methods, such as JigsawNet and others (Pirrone et al., 2021; Zhang et al., 2022a), exhibit quadratic time complexity due to exhaustive pair iteration and rarely address imperfections like stains, mold spots, or contour defects.

To address the aforementioned shortcomings, we propose ShreddingNet, a coarse-to-fine two-stage pipeline for multi-source manuscript restoration. The algorithm is divided into two stages: coarse matching and fine-grained matching. In the coarse matching stage, contour-texture bimodal features are first extracted for each fragment and fused. Based on these features, a similarity matrix between fragments is computed. For each fragment, the Top-K fragments with the highest similarity scores are selected as candidate matching pairs, and fragments from the same source are clustered according to the matching relationships at this stage. In the fine-grained matching stage, a decoder with an inter-fragment feature interaction mechanism is employed to further evaluate the candidate matching pairs for each fragment, yielding final matching scores and the local transformation matrices for the matching pairs. After eliminating low-scoring matching pairs, a best-first algorithm is used to obtain the global transformation matrices for all fragments, which are then utilized for splicing. This algorithm is capable of multi-source manuscript restoration tasks with high accuracy, exhibiting robustness to potential issues such as different image types, stains, mold spots, and contour defects on the fragments. Furthermore, the algorithm reduces the time complexity measured by the number of model invocations to a linear $O(n)$ complexity with respect to the number of fragments. The main contributions of our work are as follows:

1. Our work addresses the multi-source manuscript restoration task without any restrictive conditions or strong assumptions. The algorithm leverages the property that erroneous matching pairs in the candidate set obtained through coarse matching rarely involve fragments from different source images, enabling high-precision clustering to separate fragments belonging to distinct source images.

2. Our work reduces the number of candidate fragment pairs to a linear relationship with the number of fragments through coarse matching, thereby lowering the time complexity, measured by the number of model invocations, to a linear relationship with the number of fragments.

3. In experiments across various image quantities on two datasets, the algorithm achieves a global assembly F1-score consistently above 95%, representing the current state-of-the-art result. For robustness, experiments demonstrate that even under conditions of severe fragment degradation with simultaneous presence of stains, mold spots, and contour defects, the algorithm can still achieve a global assembly F1-score ranging from 85.53% to 88.58%.

## 2 RELATED WORKS

### 2.1 FRAGMENT FEATURE EXTRACTION

Prior innovations have primarily focused on two aspects: fragment feature extraction and pairwise fragment matching. Regarding fragment feature extraction, early studies predominantly concentrate on geometric features (Derech et al., 2021; Guo et al., 2016; Huang et al., 2006; Kong & Kimia, 2001; Mikolajczyk & Schmid, 2005; Panagiotakis et al., 2022; Savino & Tonazzini, 2016) or rule-based algorithms (da Gama Leitao & Stolfi, 2002a; Hui et al., 2013; Tsamoura & Pitas, 2010a; Yang et al., 2017; Zhang & Li, 2014; Zhang et al., 2022b). Additionally, subsequent works introduce a polar coordinate discretization method, transforming contour features into time series (da Gama Leitao & Stolfi, 2002b; Rusinkiewicz & Levoy, 2001; Schenker, 1992; Tsamoura & Pitas, 2010b; Zhang et al., 2022a) to achieve improved matching performance. Later advancements adopt learning-based approaches (Funkhouser et al., 2011a; Hossieni et al., 2023; Markaki & Panagiotakis, 2023; Paumard et al., 2018a;b; Richter et al., 2013) like CNN architectures (Zheng et al., 2024) or text recognition priors (Pengcheng et al., 2017; Zhang et al., 2023) for feature extraction. Furthermore, approaches based on fragment patches (Zhou et al., 2025) effectively address the issue of significant size disparities in complex textured fragments by extracting patches near fragment contour points.

### 2.2 PAIRWISE FRAGMENT MATCHING

In the context of pairwise fragment matching, the Siamese architecture with VGG (Simonyan, 2014) or ResNet (He et al., 2016) backbone networks (Pirrone et al., 2021; Zhang et al., 2022a) computes the similarity of fragment pairs using two parameter-shared feature extraction modules. To address the limitations of traditional rule-based classifiers, researchers (Le & Li, 2019) introduce modules such as CNN classifiers at the backend of traditional classifiers, constructing a cascaded architecture of "initial

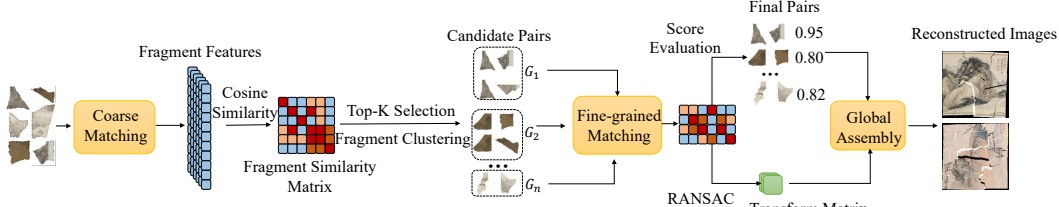

Figure 1: Overall architecture of our algorithm. We obtain candidate matching pairs and perform fragment clustering through the coarse matching stage, followed by determining the final matching pairs, their matching scores, and relative transformation matrices through the fine-grained matching stage.

screening by traditional classifiers + neural network correction." Additionally, some studies (Zhang et al., 2025) explicitly models inter-fragment relationships by combining an optimal transport layer with graph neural networks (Hamilton et al., 2017; Kipf & Welling, 2016), enabling the elimination of outlier fragments while maintaining matching accuracy.

### 2.3 DISCUSSION WITH OTHER MANUSCRIPT RESTORATION PIPELINES

Previous works have several shortcomings. Firstly, few studies address multi-source manuscript restoration or fragment degradation issues, and most rely solely on the contour features of fragments. The use of Siamese frameworks requires models to iterate through all possible fragment pairs, resulting in high time complexity. Meanwhile, algorithms that avoid iterating through all fragment pairs often face limitations, such as requiring fragments to contain text (Assael et al., 2022; Zhang et al., 2023) or a fixed number of outlier fragments (Zhang et al., 2025). Additionally, many studies utilize non-open-source datasets (Abitbol et al., 2021; Pirrone et al., 2021; Zheng et al., 2024) and lack publicly available source codes, making reproduction challenging.

In our approach, we support multi-source manuscript restoration through a coarse-to-fine two-stage matching pipeline. The coarse matching stage compares individual fragment features to filter a small set of candidate fragment pairs from all possible pairs, addressing the issue of quadratic time complexity measured by model invocations. Furthermore, we account for the impact of fragment degradation issues, such as stains, mold spots, and contour defects, and demonstrate that our method is robust to these challenges.

## 3 METHODOLOGY

### 3.1 ARCHITECTURE OVERVIEW

The overall architecture of our proposed algorithm is illustrated in Figure 1. The formulation of the problem addressed by this algorithm is provided in Appendix B. For the dataset, due to the scarcity of publicly available datasets and the difficulty of collecting large-scale real-world manuscript fragments, we adopt an algorithm-based synthetic fragment dataset generation method proposed in PairingNet to simulate the morphology of fragments resulting from real manuscript damage.

In the coarse matching stage, feature vectors are extracted from $B$ input fragments, and cosine similarity is used to compute a similarity matrix. For each fragment, the Top-$K$ most similar fragments are selected as candidate pairs, reducing potential matches from $B^2$ to under $B \times K$. The fine-grained matching stage refines these candidates by calculating a contour point matching matrix for each pair, filtering out low-scoring pairs, and computing local transformation matrices to splice final matches into larger fragments. Finally, in the global assembly stage, a best-first algorithm constructs a splicing spanning tree for each source image, which is traversed to reconstruct all images.

### 3.2 COARSE MATCHING

The coarse matching stage, as the initial phase of the proposed algorithm, aims to generate a number of candidate matching pairs that scale linearly with the number of fragments. This stage retains most

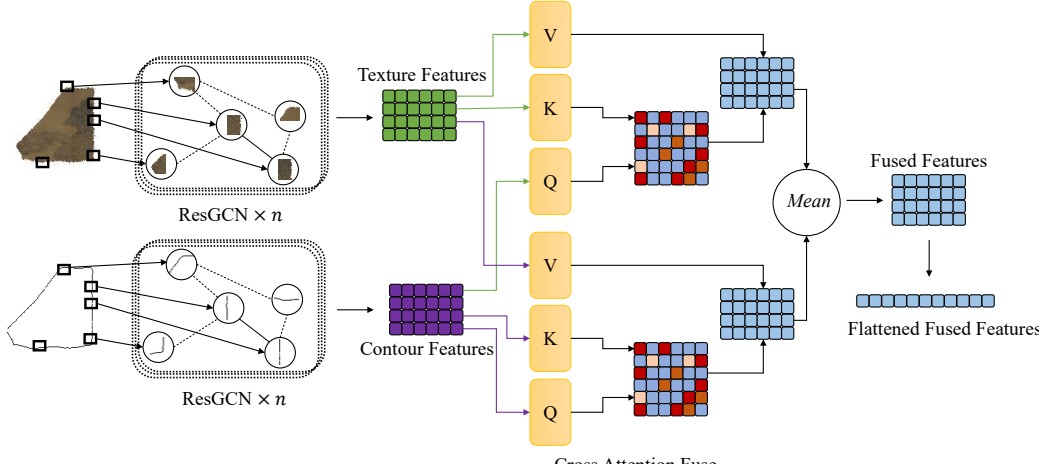

Figure 2: Architecture of the coarse matching stage. The input consists of individual fragments. The texture-contour bimodal features of each fragment are extracted and fused.

of the correct matching pairs while ensuring that erroneous matching pairs from different images are minimal. The coarse matching pipeline, as illustrated in Figure 2, primarily comprises two core modules: contour-texture bimodal feature extraction and cross-modal feature fusion.

Fragments, unlike complete images, vary significantly in size, orientation, and shape, often lacking clear semantic information. This variability complicates direct image input, leading to inconsistent feature scales, orientation-sensitive features, and reduced expressive capacity due to noise. For fragment inputs, our approach extracts each fragment's contour and generates $7 \times 7$ patches centered at each contour point, sourced from both the original image and the contour binary map(the extracted contour). These bimodal patches form a sequence, $I \in \mathbb{R}^{L \times 7 \times 7}$. The number of contour points varies across fragments, as longer contours naturally have more points. We impose a adjustable maximum limit $L$ on contour points. Contour points exceeding $L$ are truncated (in all experiments, we ensure that truncation does not occur by adjusting $L$).

For feature extraction, following the previous works (Zhang et al., 2025; Zhou et al., 2025), we employ a multilayer ResGCN (Li et al., 2019) network. The texture and contour patch sequences are transformed into graph structures using the nearest-neighbor method, with edges connecting each patch to its eight nearest neighbors. These graphs are processed by two separate 14-layer ResGCN models with non-shared parameters, yielding texture and contour feature matrices, $F_t, F_c \in \mathbb{R}^{L \times dim}$, where $dim$ is the feature vector dimension.

For feature fusion, to achieve deep integration of texture and contour features, we introduce a cross-attention mechanism to facilitate cross-modal interaction. Compared to simple feature concatenation or weighted summation, cross-attention can automatically focus on feature regions with strong correlations between the two modalities. Specifically, the texture features are first used as the Query, with contour features serving as the Key and Value, to obtain enhanced texture features. Subsequently, the contour features are used as the Query, with texture features as the Key and Value, to derive enhanced contour features. The average of the enhanced texture and contour features is taken as the fused feature. In summary, the fused feature is given by Equation 1:

$$F_{fused} = (Att(F_t, F_c, F_c) + Att(F_c, F_t, F_t))/2, \tag{1}$$

where $F_t \in \mathbb{R}^{L \times dim}$ represents the texture features, $F_c \in \mathbb{R}^{L \times dim}$ represents the contour features, and $Att(Q, K, V)$ denotes the cross-attention mechanism. $Q$, $K$ and $V$ respectively represent Query, Key, and Value. Finally, we flatten $F_{fused}$ to serve as the feature vector for the fragment.

Following the aforementioned processing pipeline, we compute the cosine similarity between fragment feature vectors to select the Top-K(K as a hyperparameter) candidate pairs with the highest similarity for each fragment. Since the fragments originate from multiple source images, it is necessary to partition the fragment set such that fragments from the same source image are grouped into a

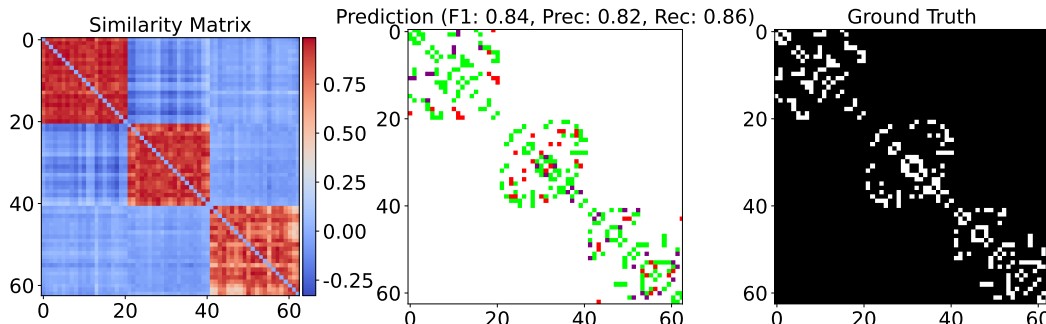

Figure 3: The results of the coarse-to-fine two-stage matching algorithm. The left heatmap shows similarity scores from the coarse matching stage, achieving high recall but low precision, with most erroneous matches from the same source image. The middle image displays fine-grained matching results (green for correct, red for incorrect and purple for missed pairs), and the right image presents ground-truth matches.

single subset. Here we have a large number of candidate matching pairs. Although these candidate pairs have low precision, most erroneous matches are between fragments from the same source image, as shown in Figure 3.

Therefore, we leverage these candidate matching pairs for clustering, grouping matching pairs from the same source image into a single cluster and eliminating erroneous matching pairs between fragments from different sources. At this point, the candidate matching pairs can be represented as a graph structure, where each fragment corresponds to a node, and each matching pair corresponds to an edge. The objective is to cluster fragments from the same source image together while keeping fragments from different source images separate. This is a typical graph clustering problem, where the cluster label of a node is determined by the labels of its neighboring nodes. We adopt the label propagation algorithm (LPA) for graph clustering since LPA does not require a predefined number of clusters and has low time complexity. Other clustering methods, including K-Means and spectral clustering, suffer from limitations such as incompatibility with graph data structures, mandatory pre-specification of cluster numbers, and high time complexity. LPA treats graph nodes as basic units, initially assigning each node a unique label. Labels are then propagated and merged through local neighborhood information, enabling node clustering within the graph.

### 3.3 FINE-GRAINED MATCHING

In the coarse matching stage, the algorithm achieves high-recall candidate matching pairs, but the candidate set contains a significant number of erroneous matches. In the fine-grained matching stage, as illustrated in Figure 4, an inter-fragment feature interaction mechanism is employed to enhance matching precision, filter out the final matching pairs, and compute the transformation matrices for these final pairs.

Unlike the coarse matching stage, which processes individual fragments, the fine-grained matching stage takes fragment pairs as input. This enables the network to perform more accurate matching by integrating information from both fragments through the inter-fragment feature interaction. We employ a decoder composed of stacked cross-attention layers to facilitate this interaction. For an input fragment pair obtained from the coarse matching stage, two feature matrices are derived: $f_1^1 \in \mathbb{R}^{L_1 \times dim}$ and $f_2^1 \in \mathbb{R}^{L_2 \times dim}$, where $L_1$ and $L_2$ represent the number of contour points for the two fragments, respectively, and $dim$ denotes the feature dimension of the contour points. Ultimately, the interacted feature matrices are multiplied to obtain the contour point matching matrix $M \in \mathbb{R}^{L_1 \times L_2}$. The computation of $M$ is given by Equation 2:

$$
\begin{aligned}
f_1^i &= Att(f_2^{i-1}, f_1^{i-1}, f_1^{i-1}) \qquad (i \in [1, n]), \\
f_2^i &= Att(f_1^{i-1}, f_2^{i-1}, f_2^{i-1}) \qquad (i \in [1, n]), \\
M &= f_1^n \times (f_2^n)^T.
\end{aligned}
\tag{2}
$$

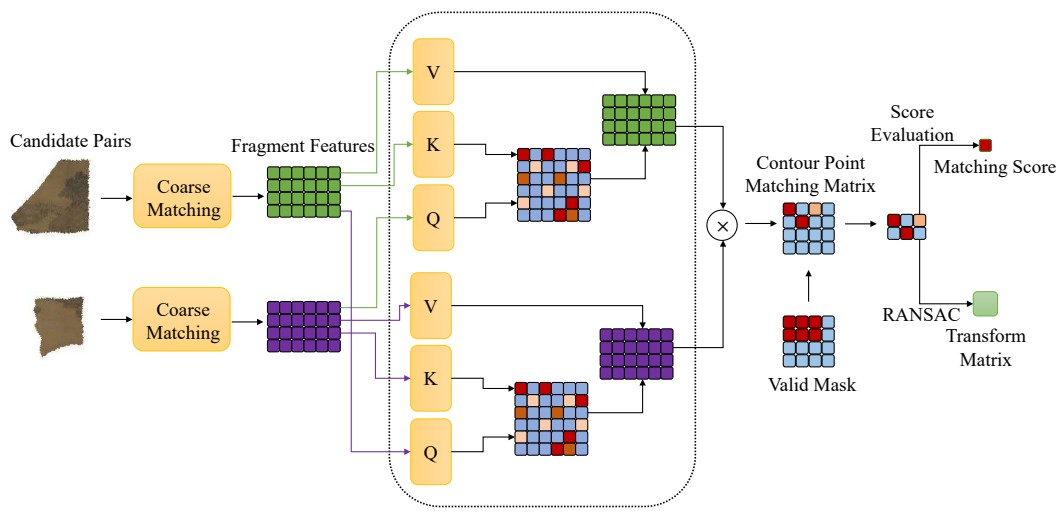

Figure 4: Architecture of the fine-grained matching stage. Fragment pairs obtain a contour point matching matrix through inter-fragment feature interaction. This matrix is processed by applying a valid mask to remove meaningless values beyond each fragment's contour, followed by score evaluation to obtain matching scores and final matching pairs.

where $f_1^i$ and $f_2^i$ are the feature matrices after the $i$-th layer of interaction, $M$ is the matching relationship matrix, $Att(\cdot)$ represents the attention layer, and $n$ is the number of stacked layers.

In the contour point matching matrix, contour points that align well after splicing exhibit high similarity, while non-aligning points show low similarity. The length of continuous high-similarity diagonal lines in the matrix serves as a key indicator for evaluating matching pairs. When two fragments are correctly matched, the contour points along their fractured edges should exhibit a strict one-to-one correspondence, resulting in a continuous high-similarity region along the main diagonal of the matrix and vice versa.

A simple idea is that the matching probability of a pair can be assessed by calculating the length of continuous high-similarity diagonals in the contour point matching matrix. However, due to errors in the upstream models, the matching matrix may contain noise, leading to discontinuities in the diagonal or interference from noise on either side of the diagonal. Rule-based methods that simply compute the longest diagonal length struggle to handle these cases flexibly, resulting in significant errors in matching probability estimation, as shown in Figure 14 in Appendix H.

To compute matching scores, we employ a lightweight network comprising three CNN layers to capture patterns in the similarity matrix, such as high-similarity diagonals, followed by an MLP that maps features to scores via a Sigmoid function. This approach efficiently learns complex patterns like diagonal continuity and density. Scores above a threshold(as a hyperparameter) determine the final matching pairs, for which we compute local transformation matrices using RANSAC (Fischler & Bolles, 1981), following PairingNet.

The global assembly stage integrates fragments by deriving global transformation matrices from final matching pairs, which include local transformation matrices. These transformations map fragments into a unified coordinate system to reconstruct the complete image. The final matching pairs form a graph, where the nodes and edges represent fragments and fragment pairs, respectively. The edge weights consist of the matching scores and local transformation matrices of the fragment pairs. Each connected subgraph of this graph corresponds to a source image. The global transformation matrices for all fragments can be determined by identifying any spanning tree within each connected subgraph. We select the maximum spanning tree for each connected subgraph and use it to compute the global transformation matrices for all fragments.

# 4 EXPERIMENTS

## 4.1 EVALUATION SETTINGS

**Data.** We utilize synthetic datasets for algorithm training and performance evaluation, constructed in two stages: raw material collection and synthetic dataset generation. The raw materials are categorized into two types: (1) art_2192 (Xue, 2020), consisting of 2,192 digitized images of Chinese artworks and calligraphy, and (2) pex_2000, comprising 2,000 high-resolution digital photographs from kaggle public dataset (Mills, 2023). Details and examples of the two datasets are shown in Appendix D.1 and Figure 6 in Appendix E.

Using the dataset generation algorithm from Section 3.1, we create four synthetic datasets at different difficulty levels (normal and hard). In the hard dataset, we introduced common degradation conditions in the test set, such as stains, mold spots, and contour defects. No degradation conditions were added to the training set to evaluate the model's zero-shot capability in handling these conditions. Please refer to Appendix D for more details on dataset generation.

**Implementation details.** The training pipeline optimizes three networks: coarse matching, fine-grained matching, and score evaluation. All networks are trained using the Adam optimizer (Kingma & Ba, 2014) with cosine annealing learning rate decay and a weight decay of $5 \times 10^{-4}$ to prevent overfitting. The dataset is split into training, validation, and test sets (7:1:2 ratio) with a fixed random seed of 1024 for reproducibility. The loss function and implementation details of each module can be found in Appendix D.

## 4.2 RESULTS

### 4.2.1 PERFORMANCE

The performance of the proposed algorithm's coarse matching, fine-grained matching, score evaluation, and global assembly are evaluated. For an introduction to the metrics used in the experiment, please refer to Appendix C.

Test results on the art_2192 and pex_2000 datasets are shown in Table 1, with global assembly results of normal datasets in Figure 8 and Figure 7 in Appendix F. Results of hard datasets can be found in the supplementary materials. In Table 1, we conduct multiple experiments(random seeds 512, 1024, and 2048) using different random seeds and incorporate averaged results with error bars. The algorithm achieves over 95% precision and recall in global assembly for normal datasets, maintaining robust performance across image types and source image counts. On hard datasets with stains, mold spots, and contour defects, precision remains above 95%, with recall between 75–82%, demonstrating effective handling of real-world degradation.

### 4.2.2 COMPARATIVE EXPERIMENT

Few baseline methods suit multi-source manuscript restoration (Zhou et al., 2025). We compare our method against three adapted baselines(adjustments are provided in Appendix D.4): a rule-based method (Zhang & Li, 2014), JigsawNet, and PairingNet. (LLMCO4MR is not included, as it has not yet released its code, dataset, or model hyperparameter setting details, making it difficult to reproduce.) For baselines with $O(n^2)$ complexity(JigsawNet and rule-based method), testing is limited to the first three dataset batches to manage computational time. Results are shown in Table 2. The intuitive visual comparison results of the images are presented in Figure 9 in Appendix F. It is evident from the Figure 9 that ShreddingNet successfully restored nearly all images correctly, whereas the images restored by JigsawNet and PairingNet exhibit varying degrees of defects.

On art_2192 (3 images), our method achieves 77.07% fine-grained matching precision and 99.15% global assembly precision, outperforming the best baseline (PairingNet) by 13.41 and 4.34 points, respectively. The rule-based method struggles with coarse matching and fails in assembly. On pex_2000 (3 images), our method reaches 86.61% matching precision, surpassing PairingNet by 8.57 points. With 6 images, our method sustains 96.38% global assembly recall, while PairingNet drops to 88.54%, highlighting our stability in multi-source scenarios.

Table 1: Performance of each module of the proposed method across different datasets and numbers of images. The numbers on the left are mean values, and the numbers on the right are 2-sigma standard deviations. In the table, the abbreviations are defined as follows: IC stands for Image Counts, CM stands for Coarse Matching, FM stands for Fine-grained Matching, SE stands for Score Evaluation, and GA stands for Global Assembly. The following tables follow the same convention.

| Datasets | Difficulty | IC | CM | | FM | | SE | GA | |
|---|---|---|---|---|---|---|---|---|---|
| | | | Prec(%) | Rec(%) | Prec(%) | Rec(%) | AUC(%) | Prec(%) | Rec(%) |
| art_2192 | normal | 3 | $19.32_{0.05}$ | $99.37_{0.03}$ | $77.07_{0.08}$ | $78.40_{0.03}$ | $92.04_{0.02}$ | $99.15_{0.03}$ | $97.59_{0.57}$ |
| | | 6 | $18.85_{0.01}$ | $98.87_{0.05}$ | $76.77_{0.02}$ | $77.04_{0.01}$ | $91.97_{0.01}$ | $98.55_{0.52}$ | $97.28_{0.41}$ |
| | | 9 | $18.52_{0.05}$ | $98.51_{0.01}$ | $76.57_{0.03}$ | $76.67_{0.38}$ | $92.06_{0.01}$ | $98.27_{0.76}$ | $97.26_{0.70}$ |
| | | 12 | $18.20_{0.01}$ | $97.90_{0.01}$ | $76.42_{0.12}$ | $75.52_{0.19}$ | $92.09_{0.02}$ | $97.84_{0.55}$ | $97.09_{0.66}$ |
| | | 15 | $18.02_{0.01}$ | $97.73_{0.01}$ | $76.34_{0.01}$ | $74.95_{0.35}$ | $92.10_{0.02}$ | $97.63_{0.95}$ | $97.15_{0.70}$ |
| | hard | 3 | $19.16_{0.07}$ | $99.11_{0.27}$ | $86.53_{0.52}$ | $55.82_{0.46}$ | $86.48_{0.06}$ | $95.72_{0.32}$ | $81.43_{0.72}$ |
| | | 6 | $18.67_{0.03}$ | $98.46_{0.11}$ | $86.62_{0.85}$ | $54.96_{0.49}$ | $86.52_{0.22}$ | $95.59_{0.29}$ | $82.18_{0.75}$ |
| | | 9 | $18.28_{0.05}$ | $97.93_{0.28}$ | $86.49_{0.07}$ | $54.17_{0.17}$ | $86.44_{0.06}$ | $95.34_{0.36}$ | $81.88_{0.36}$ |
| | | 12 | $17.96_{0.03}$ | $97.29_{0.21}$ | $86.16_{0.77}$ | $54.08_{2.20}$ | $86.44_{0.20}$ | $95.15_{0.37}$ | $81.01_{1.29}$ |
| | | 15 | $17.94_{0.03}$ | $97.23_{0.16}$ | $86.18_{0.59}$ | $53.43_{0.22}$ | $86.54_{0.10}$ | $94.95_{0.05}$ | $80.95_{0.75}$ |
| pex_2000 | normal | 3 | $12.72_{0.03}$ | $98.65_{0.06}$ | $86.61_{0.02}$ | $68.69_{0.05}$ | $91.69_{0.04}$ | $98.20_{0.05}$ | $96.78_{0.25}$ |
| | | 6 | $12.22_{0.01}$ | $97.23_{0.03}$ | $86.02_{0.01}$ | $65.92_{0.02}$ | $91.71_{0.01}$ | $97.58_{0.03}$ | $96.38_{0.11}$ |
| | | 9 | $11.93_{0.04}$ | $95.73_{0.02}$ | $85.80_{0.04}$ | $62.63_{0.05}$ | $91.82_{0.05}$ | $96.37_{0.05}$ | $95.42_{0.09}$ |
| | | 12 | $11.75_{0.01}$ | $94.96_{0.01}$ | $85.50_{0.04}$ | $61.47_{0.61}$ | $91.79_{0.05}$ | $96.07_{0.09}$ | $95.42_{0.16}$ |
| | | 15 | $11.57_{0.02}$ | $93.86_{0.06}$ | $85.36_{0.07}$ | $60.07_{0.01}$ | $91.83_{0.00}$ | $95.80_{0.08}$ | $95.01_{0.05}$ |
| | hard | 3 | $12.55_{0.01}$ | $97.84_{0.10}$ | $91.23_{0.42}$ | $48.04_{0.59}$ | $85.91_{0.08}$ | $96.94_{0.10}$ | $78.44_{0.51}$ |
| | | 6 | $12.01_{0.02}$ | $96.02_{0.16}$ | $91.18_{0.62}$ | $46.46_{0.24}$ | $85.95_{0.31}$ | $96.51_{0.70}$ | $78.56_{0.75}$ |
| | | 9 | $11.70_{0.02}$ | $94.40_{0.12}$ | $90.49_{0.12}$ | $44.14_{0.36}$ | $86.16_{0.19}$ | $95.69_{0.13}$ | $77.00_{0.47}$ |
| | | 12 | $11.50_{0.01}$ | $93.30_{0.10}$ | $90.47_{0.35}$ | $43.16_{0.67}$ | $86.09_{0.17}$ | $95.48_{0.23}$ | $76.75_{0.78}$ |
| | | 15 | $11.31_{0.04}$ | $92.22_{0.26}$ | $90.19_{0.32}$ | $42.05_{0.57}$ | $86.30_{0.14}$ | $95.03_{0.54}$ | $76.02_{0.49}$ |

Table 2: Performance comparison of the proposed method and three baseline models.

| Datasets | IC | Baseline | FM | | GA | |
|---|---|---|---|---|---|---|
| | | | Prec(%) | Rec(%) | Prec(%) | Rec(%) |
| art_2192 | 3 | Rule Based | 7.48 | 44.90 | N/A | N/A |
| | | JigsawNet | 44.55 | 69.23 | 92.79 | 82.88 |
| | | PairingNet | 63.66 | **79.09** | 94.81 | 90.49 |
| | | Ours | **77.07** | 78.40 | **99.15** | **97.59** |
| | 6 | Rule Based | 3.75 | 47.37 | N/A | N/A |
| | | JigsawNet | 43.35 | 67.66 | 92.46 | 82.28 |
| | | PairingNet | 62.86 | **78.69** | 94.10 | 90.67 |
| | | Ours | **76.77** | 77.04 | **98.55** | **97.28** |
| pex_2000 | 3 | Rule Based | 6.32 | 43.87 | N/A | N/A |
| | | JigsawNet | 41.84 | 60.27 | 88.15 | 79.21 |
| | | PairingNet | 78.04 | 66.61 | 94.83 | 88.71 |
| | | Ours | **86.61** | **68.69** | **98.20** | **96.78** |
| | 6 | Rule Based | 2.64 | 44.6 | N/A | N/A |
| | | JigsawNet | 41.26 | 58.51 | 87.09 | 72.42 |
| | | PairingNet | 77.30 | **65.96** | 94.4 | 88.54 |
| | | Ours | **86.02** | 65.92 | **97.58** | **96.38** |

To validate our method's efficiency, we compare its inference time with JigsawNet and PairingNet on the art_2192 dataset (3 images per batch), as shown in Figure 5 (left). Our method has the shortest average per-batch inference time compared to the baselines. This is because JigsawNet, which requires iterating through all possible fragment pairs, has a time complexity of $O(n^2)$, while PairingNet, despite having a time complexity of $O(n)$, incurs high computational costs due to the absence of fine-grained matching, as all candidate pairs must undergo time-consuming RANSAC computations. In contrast, our proposed method significantly reduces the number of final matching pairs through a coarse-to-fine pipeline, thereby lowering the time cost. Additionally, Figure 5 (right) demonstrates our method's linear time scaling with fragment count, given an approximately constant

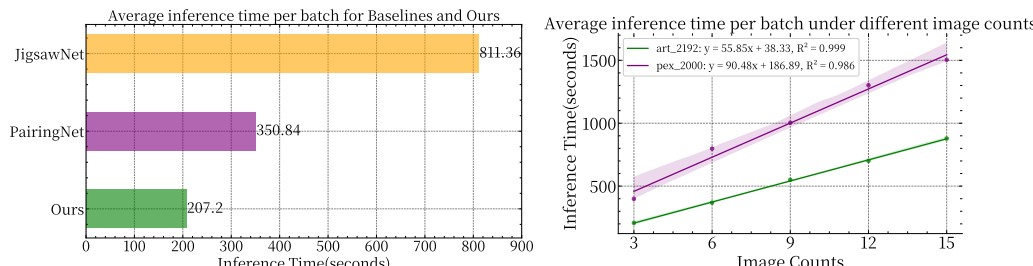

Figure 5: The left figure presents a comparison of the inference time of the proposed method with baselines on art_2192(3 images). The right figure illustrates the variation in inference time of the proposed method with respect to the number of images, conducted on both datasets. The shaded area represents the confidence interval (95% confidence level) of the regression line.

number of fragments per image. Due to the higher resolution and greater number of contour points in the pex_2000 dataset, the inference time is longer compared to the art_2192 dataset.

### 4.3 Robustness and real-world case study

To evaluate the robustness and generalizability of our proposed method, we conducted experiments on the impact of stains, molds, and contour defects, as well as real-world case studies. Detailed results and visualizations are provided in Appendix G.

We also conducted experiments on hyperparameter selection, sensitivity analysis, and module architecture choices, including the selection of K, different fragment feature extraction methods, the impact of content similarity of source images on model performance and sensitivity to orientation and patch size. The results of these experiments are presented in Appendix I.

### 4.4 Ablation study

To assess the necessity of our method's components, we performed ablation studies on the art_2192 dataset, individually omitting fragment clustering, fine-grained matching, and CNN-based score evaluation while maintaining consistent parameters. Results are detailed in Table 5 in Appendix H and Figure 13 in Appendix H. We use the Adjusted Rand Index (ARI) to evaluate whether each model can effectively separate fragments from different sources. For a detailed description of ARI, please refer to Appendix C. ARI for clustering is computed post-assembly for methods without explicit clustering.

Excluding fragment clustering lowers the ARI to 53.44%, a 38.85-point drop, underscoring its role in fragment clustering. Global assembly precision and recall fall by 1.21 and 0.28 points, respectively, due to increased erroneous matches. Without fine-grained matching, precision drops to 20.77% while recall rises to 98.49%, reflecting poor error filtering; global assembly metrics are untestable due to missing transformation matrices. Replacing CNN-based scoring with a rule-based approach reduces global assembly recall by 4.05 points, as shown in Figure 14 in appendix H, where rule-based scoring lacks the CNN's clear decision boundary.

## 5 Conclusion

In conclusion, this paper proposes ShreddingNet, a two-stage manuscript restoration network for multi-source shredded artworks. The network achieves high-accuracy splicing of multi-source fragments through a coarse-to-fine two-stage pipeline. We obtained candidate matching pairs that enable easy clustering of fragments from different source images in the coarse matching stage, thereby addressing the multi-source manuscript resolution task. In addition to solving the multi-source manuscript resolution task, this work achieves linear time complexity by ensuring the number of candidate matching pairs scales linearly with the number of fragments. Experiments demonstrate that this method is robust to common fragment degradation scenarios.

## Reproducibility statement

To facilitate researchers in reproducing the experimental results presented in this paper, we include a reproducibility statement section following the main text, detailing the experimental environment, hyperparameter settings, datasets, experimental results, and methods for accessing the code and model parameters. The experimental hardware and software environment, as well as the hyperparameter settings, are comprehensively described in 4.1 of the main text and Appendix D. For the datasets, we have hosted them on HuggingFace and provided their anonymous download links in our supplementary materials. Due to size constraints, only partial experimental results are presented in the supplementary materials. However, we have included a detailed tutorial in the supplementary materials to enable readers to reproduce all experimental results. The complete code and trained model parameters are also available in the supplementary materials, with further details provided in the *ReadMe.md* file included therein.

## Ethics statement

This work proposes an algorithm for multi-source manuscript restoration. Due to the algorithm's focus on the manuscript restoration task of art calligraphy and painting, and the fact that all datasets used are open-source and free of intellectual property disputes (see Section 4.1 for details), there are no foreseeable negative ethical issues.

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

## A  OVERVIEW

The appendix consists of 10 sections, as outlined below:

1. **Overview**: This section provides a comprehensive introduction to the appendix, summarizing the main content of each section.

2. **Problem Formulation**: This section mathematically defines the task of multi-source manuscript restoration.

3. **Metrics**: This section introduces the definitions and computational formulas of various evaluation metrics used in the experiments.

4. **Experiment Details**: This section describes the experimental details, including the settings used in dataset generation, model architecture, and the values of various hyperparameters used during training and testing.

5. **Generated Dataset Examples**: This section presents example images of fragments generated by the dataset generation algorithm.

6. **Visual Results**: This section showcases the visual results of our proposed algorithm and provides a visual comparison with other baseline methods.

7. **Robustness and Real-World Case Study**: This section presents experimental results on the algorithm's robustness to varying degrees of contour degradation and conducts a case study in a real-world setting.

8. **Ablation Study**: This section details the experimental setup, result tables, and visual outcomes of the ablation study.

9. **Additional Experiments**: This section presents additional experiments, including the selection of K, different fragment feature extraction methods, the impact of content similarity of source images on model performance and sensitivity to orientation and patch size.

10. **The Use of Large Language Models (LLMs)**: This section describes how LLMs are utilized to assist in the writing of this paper.

## B  PROBLEM FORMULATION

Suppose some original images, after physical segmentation or natural damage, form a discrete fragment set, where each fragment can be attributed to a specific source image. Let the complete fragment set be denoted as $\{V_n^m | n \in \{1, 2, \cdots, N\}, m \in \{1, 2, \cdots, M_n\}\}$, where the subscript index $n \in [1, N]$ indicates the source image to which a fragment belongs, and the superscript $m \in [1, M_n]$ represents the number of fragments associated with that image.

The algorithm is required to output a set of weighted connected graphs $\{G_1(V_1, E_1, T_1), \ldots, G_N(V_N, E_N, T_N)\}$, where each connected graph $G_i$ corresponds to the global assembly process for the $i$-th original image. Each connected graph must satisfy a tree structure constraint. The vertex set $V_i$ of $G_i$ forms a strict partition subset of the original fragment set, the edge set $E_i$ contains edges $(u_j, v_j)$ representing detected fragment pairs, and the edge weight $t_j \in T_i$ denotes the local affine transformation matrix required to align fragment $u_j$ to $v_j$.

## C  METRICS

This section describes the computation details of the evaluation metrics mentioned in 4.2.1. In our experiments, Precision and recall assess matching modules, Adjusted Rand Index (ARI) evaluates clustering in the ablation study, Area Under the Curve (AUC) measures score evaluation. Global assembly performance uses edge-based precision and recall.

Precision measures the proportion of true matching pairs among the screened candidate pairs, calculated as $Prec = \frac{TP}{TP+FP}$, where $TP$ represents the number of correctly identified true matching pairs, and $FP$ denotes the number of falsely identified matching pairs. Recall reflects the proportion of true matching pairs successfully screened, computed as $Rec = \frac{TP}{TP+FN}$, where $FN$ indicates the number of true matching pairs that were not correctly screened.

For evaluating fragment clustering performance in the ablation study, we adopt the Adjusted Rand Index (ARI). The original Rand Index (RI) measures the consistency of sample pairs between clustering results and true partitions, calculated as $RI = \frac{TP+TN}{TP+FP+FN+TN}$, where $TP$ is the number of true positive pairs, and $TN$ is the number of true negative pairs. The Adjusted Rand Index improves upon this by adjusting for chance, with its formula given in Equation 3:

$$ARI = \frac{RI - E[RI]}{\max(RI) - E[RI]}..\tag{3}$$

where $E[RI]$ is the expected value of RI. ARI ranges from $[-1, 1]$, with $ARI = 1$ indicating perfect agreement between clustering results and true partitions, and $ARI \approx 0$ for random partitions.

For evaluating score evaluation, we use the Area Under the Curve (AUC) as the metric. AUC quantifies the model's ability to rank positive and negative samples by calculating the area under the receiver operating characteristic curve, with values ranging from 0.5 to 1.

For the global assembly stage, the evaluation focuses on the geometric rationality of the overall spliced structure, ensuring no cracks or overlapping regions. Building upon the metrics defined in JigsawNet, we propose edge-based precision and recall calculations. Global precision is defined as the proportion of true matching pairs among the pairs selected during maximum spanning tree construction, as given in Equation 4:

$$Prec_{global} = \frac{|E_{correct}|}{|E_{selected}|}..\tag{4}$$

where $E_{correct}$ is the set of correct matching pairs, and $E_{selected}$ is the set of pairs selected by the algorithm. Global recall is defined based on geometric error criteria, calculating the proportion of true matching pairs with a rotation error less than $5°$ and a translation error less than 100 pixels (Le & Li, 2019), as shown in Equation 5:

$$Rec_{global} = \frac{|E_{valid}|}{|E_{truth}|}..\tag{5}$$

where $E_{valid}$ is the set of pairs passing the error criteria, and $E_{truth}$ is the set of all true matching pairs. This edge-based metric, unlike JigsawNet's node-based recall, focuses on relative fragment positions, unaffected by global translation or rotation.

## D    EXPERIMENT DETAILS

### D.1    DATASET GENERATION

The art_2192 dataset (Xue, 2020) consists of 2,192 high-quality traditional Chinese landscape paintings. All paintings are sized $512 \times 512$, from the four sources: Princeton University Art Museum, Harvard University Art Museum, Metropolitan Museum of Art and Smithsonian's Freer Gallery of Art. The pex_2000 dataset (Mills, 2023) contains resized and cropped free-use stock photos from Pexels. All the images have minimum dimensions of 768p and maximum dimensions that are multiples of 32.

For fragment details, each original image is segmented into 20–40 fragments to simulate damage. The hard dataset incorporates simulated stains, mold spots, and contour defects to assess robustness. Stains are modeled as dark circular spots, with radii $\sim \mathcal{N}(0, 5)$ pixels, and 10–15 stains per fragment. Mold spots use irregular green-to-brown textures, generated with Perlin noise. Contour defects involve retracting contour segments (2–30 pixels) along the normal direction by 1–5 pixels with a 30% probability per fragment, simulating contour corrosion.

### D.2    ARCHITECTURE DETAILS

All hyperparameters of the multilayer ResGCN network in the coarse matching stage are identical to those in PairingNet. The feature vector dimension $dim$ is 64. In the fine-grained matching stage, we

Table 3: Compute resources for the experiment.

| Configuration category | Description |
| --- | --- |
| **Software** | |
| OS | Ubuntu 18.04.3 LTS |
| Programming language | Python 3.10.15 |
| Framework | PyTorch 2.5.1 + CUDA 12.2 |
| **Hardware** | |
| CPU | Intel(R) Xeon(R) Silver 4210 CPU @ 2.20GHz |
| GPU | NVIDIA TITAN RTX $\times$ 1 (24576 MiB) |
| Memory | DDR4 251GB |
| Capacity | 7.3TB |

stacked 2 layers of Cross Attention Decoder ($n = 2$ in Figure 4), with each decoder containing 8 attention heads.

For all experiments conducted on the art_2192 dataset, we set the maximum number of contour points $L$ to 1280. For all experiments conducted on the pex_2000 dataset, we set $L$ to 2400. These settings ensure that no fragment is truncated due to an excessively long contour.

### D.3 TRAINING DETAILS

The coarse matching network is trained on true matching fragment pairs using InfoNCE (Dai & Lin, 2017) loss (temperature coefficient 0.12), with an initial learning rate of $10^{-4}$, batch size of 75, and 128 epochs. The fine-grained matching network uses the same data but employs FocalLoss (Li & Harada, 2022) ($\alpha = 0.55$, $\gamma = 8$), with an initial learning rate of $10^{-3}$, batch size of 54, and 128 epochs. Both processes take approximately 5 days. The score evaluation network is trained on balanced positive and negative sample pairs (negative samples from random non-matching pairs of the same source image) using binary cross-entropy loss, with an initial learning rate of $10^{-4}$, batch size of 20, and 10 epochs, lasting about 1 day.

Training and testing are conducted on a single NVIDIA TITAN RTX with 24 GB memory. More information on the compute resources used in the experiment is shown in Table 3

### D.4 TESTING DETAILS

We select the checkpoint with the lowest validation loss for testing. The test set consists of fragment data independent of the training and validation sets, accounting for 20% of the total data. The generation process for the test set follows the same synthetic strategy as the training set to ensure distributional consistency. All tests use a fixed random seed of 1024 to guarantee reproducible results.

We employed different testing hyperparameters for the test sets of different datasets. For the art_2192 dataset, in the coarse matching stage, $K$ was set to 20, meaning 20 fragments were selected for each fragment to form candidate matching pairs. In the fine-grained matching stage, the score evaluation threshold was set to 0.5. For the pex_2000 dataset, in the coarse matching stage, $K$ was set to 30, meaning 30 fragments were selected for each fragment to form candidate matching pairs. In the fine-grained matching stage, the score evaluation threshold was also set to 0.5.

In comparative experiments, we made minor adjustments to the baseline methods to enable them to perform multi-source manuscript restoration tasks. For JigsawNet, since its proposed HLM global assembly algorithm tends to splice all fragments into a single image, which is unsuitable for multi-source manuscript restoration, we replaced it with the global assembly module used in this work. For PairingNet, we queried the Top-$K$ adjacent fragments for each fragment, using the same $K$ values as in our proposed algorithm. As matching scores for fragment pairs could not be obtained, in the global assembly stage, we arbitrarily selected a spanning tree for splicing.

# E  GENERATED DATASET EXAMPLES

Here we provide some examples of synthetic datasets(shown in Figure 6), where white curves represent the generated contours, and green edges indicate the adjacency relationships between fragments.

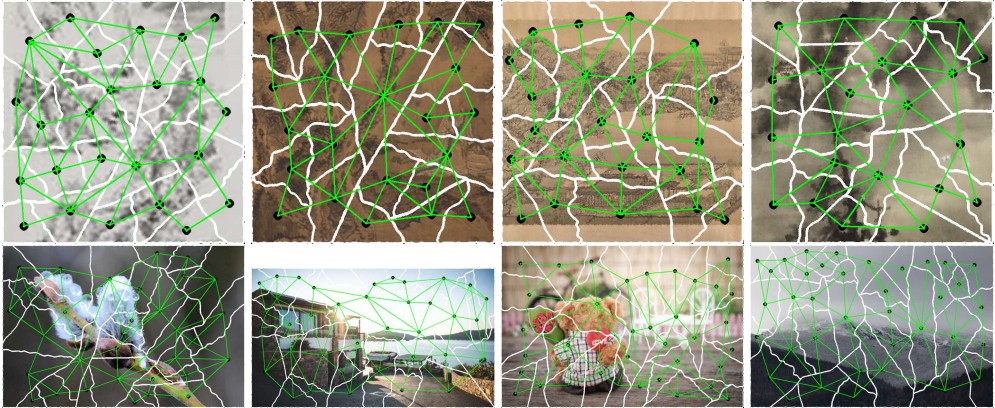

Figure 6: The fragments generated by the dataset generation algorithm are shown. The upper part of the figure displays the generation results for the art_2192 dataset, while the lower part shows the results for the pex_2000 dataset.

# F  VISUAL RESULTS

Here we present 40 assembled images (shown in Figure 7 and Figure 8). The images are randomly selected from the experimental results using a mixture of fragments generated from three images at a time from the art_2192 and pex_2000 datasets as input, which can represent the intuitive performance of our model.

Here, we provide a visual comparison of the results of ShreddingNet and baseline methods (Figure 9). We randomly selected results from three batches (each batch containing three source images) for presentation.

# G  ROBUSTNESS AND REAL-WORLD CASE STUDY

To further demonstrate the robustness and generalizability of our proposed method in real-world scenarios, we investigate the impact of varying numbers of stains, molds, and different degrees of contour defects on the model's performance. Additionally, we conduct a case study in a real-world setting.

For stains and molds, we gradually increase the total number of stains and molds from none to severe (35 spots per fragment). Throughout this process, the number of stains and molds is kept equal (even sum) or differed by 1 (odd sum). For contour defects, we progressively increase the contour defect ratio from none to severe (35%). Other settings remain consistent with the corresponding experiments in Table 1.

As shown in Figure 10, the decline in Global Assembly F1 Score remains approximately linear and relatively gradual for both stains&molds and contour defect, demonstrating the strong robustness of our method. Figure 11 intuitively illustrates the model's restoration results under varying degrees of degradation. As shown in the figure, the model maintains robust restoration performance across different levels of degradation. Furthermore, the table reveals that stains&molds have a greater adverse impact on assembly performance compared to contour defect. This is likely because stains&molds simultaneously degrade both the contour and textural features of fragments, whereas edge breakage primarily affects only the contour features.

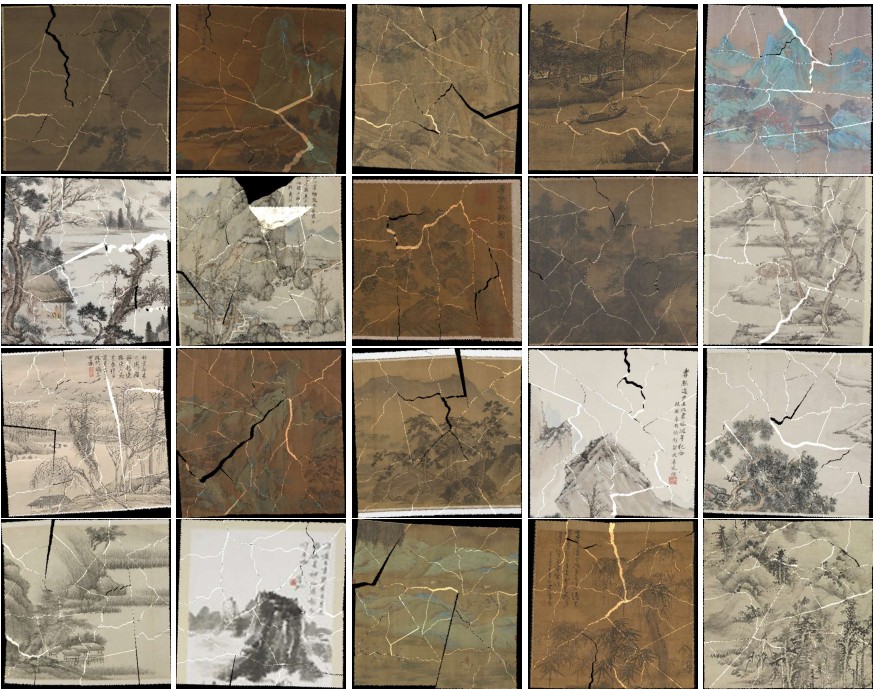

Figure 7: These 20 images are obtained from the splicing results of the art_2192 normal dataset, randomly selected from 7 batches, with each batch comprising all fragments generated from 3 complete images.

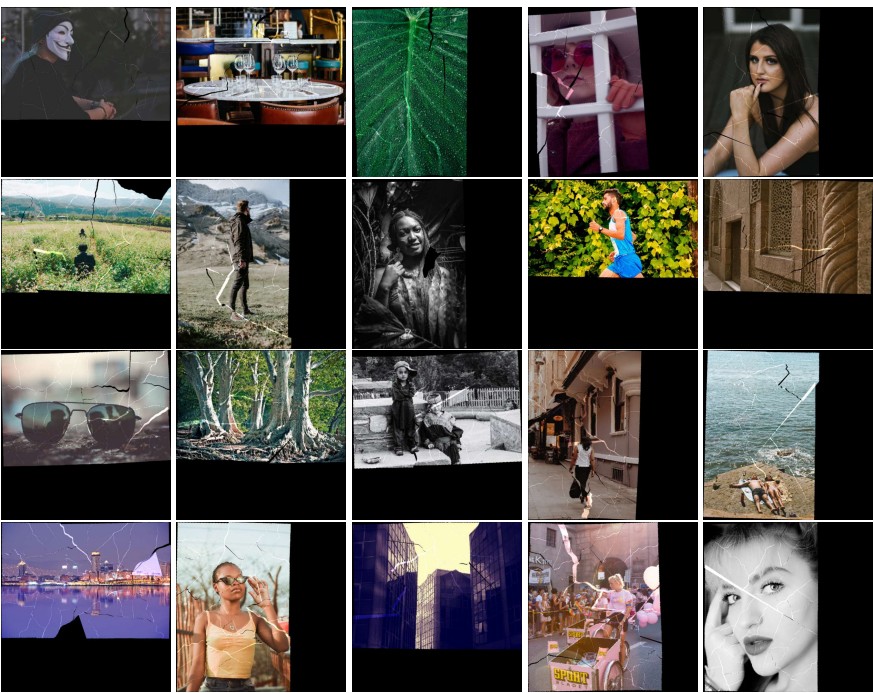

Figure 8: These 20 images are obtained from the splicing results of the pex_2000 normal dataset, randomly selected from 7 batches, with each batch comprising all fragments generated from 3 complete images.

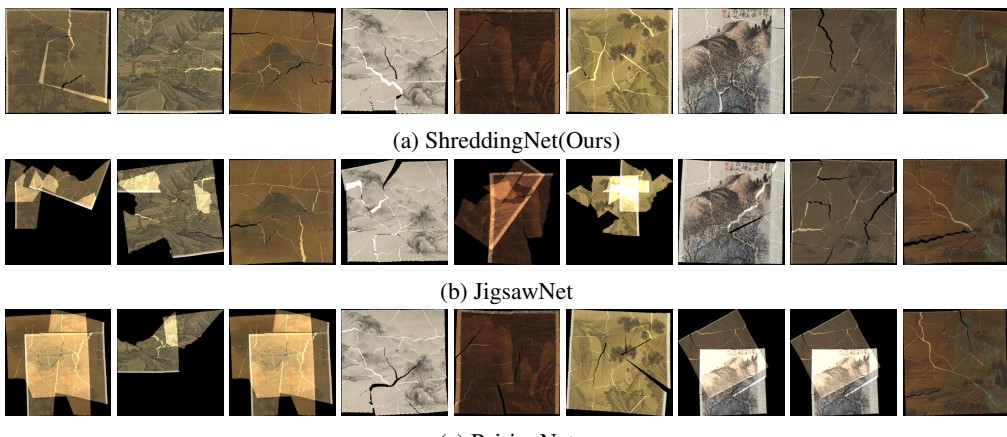

(a) ShreddingNet(Ours)

(b) JigsawNet

(c) PairingNet

Figure 9: Comparison of Results for ShreddingNet (Ours), JigsawNet, and PairingNet. The appearance of identical result images is due to the model failing to correctly classify the source images, causing multiple images to be concatenated.

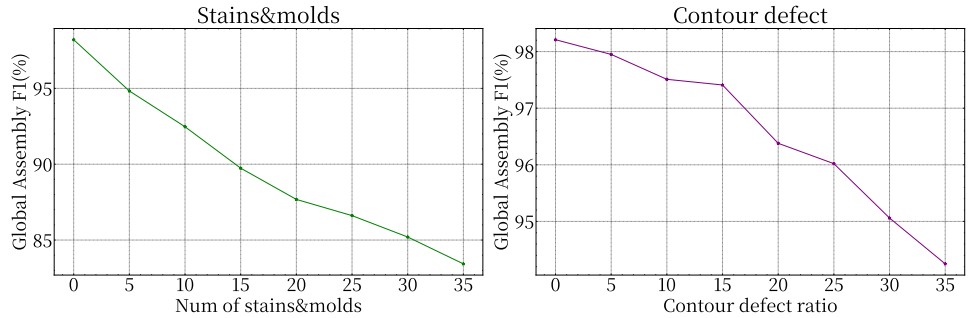

Figure 10: The left figure illustrates the impact of an increasing number of stains and molds on model performance, while the right figure shows the impact of progressively severe contour defects on model performance. In both figures, the horizontal axis represents the severity of degradation, and the vertical axis represents the Global Assembly F1 Score.

Due to the lack of a large-scale real-world fragment test set, we conduct two case studies to validate the model's performance in real-world fragment restoration scenarios. We generate fragments through the process illustrated in Figure 12. Specifically, we randomly select images not included in the model's training set, print them out, manually tore them into fragments, and mix fragments from three different source images. These fragments are then scanned into digital images, with the background removed, and use as inputs to the model. The experiments demonstrate that the model nearly completely reconstructed all the images (Refer to Figure 12.) The global assembly precision and recall of the two cases are shown in Table 4.

Due to the significant workload involved in manually generating fragments and digitally scanning all fragments, we do not conduct additional real-world case studies. If a large-scale real-world fragment dataset becomes available, we will test our model on it.

## H ABLATION STUDY

Here, we provide the result table (Table 5) of our ablation study. Please refer to 4.4 for a more detailed explanation.

We also visually demonstrate the impact of missing a specific module on the final assembly results through Figure 13. Additionally, Figure 14 illustrates the probability density distributions (obtained via Gaussian kernel density estimation) of matching scores assigned by the CNN-based score

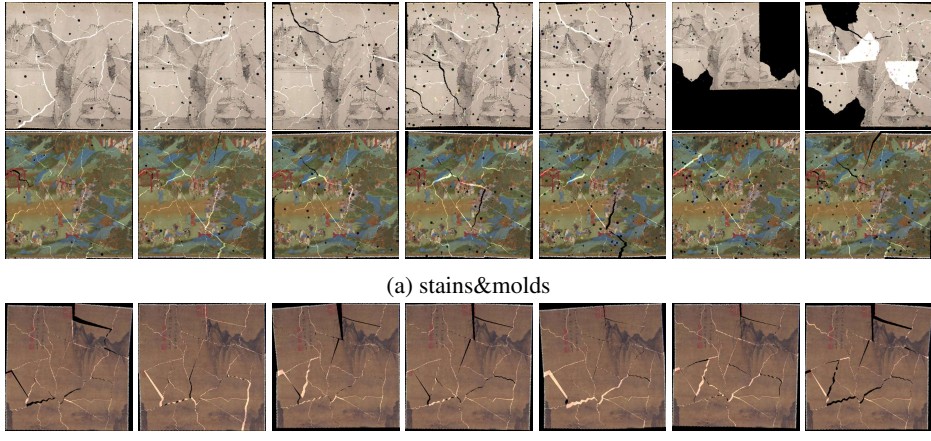

(a) stains&molds

(b) contour defect

Figure 11: This figure intuitively illustrates the model's restoration results under varying degrees of degradation. The top two rows display the model's output images when the number of stains and mold varies from 5 to 35. The bottom row shows the model's output images when the degree of contour defects varies from 5 to 35.

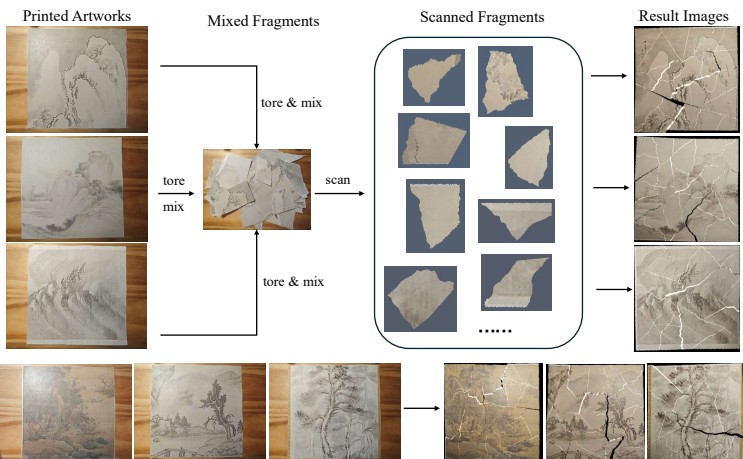

Figure 12: The figure provides the original images and results of two real-world case studies (case1 at the top and case2 at the bottom), and takes case1 as an example to describe the method of manually generating fragments.

evaluation module and the rule-based score evaluation module for correct and incorrect matching pairs. The rule-based score is not normalized, resulting in scores that may exceed 1.

## I   ADDITIONAL EXPERIMENTS

To verify the impact of the choice of K on model performance and time consumption, we perform a study on art_2192 (3 source images), varying K from 5 to 40. The results, including Global Assembly metrics and inference time, are presented in Figure 15.

The average inference time exhibits linear growth with increasing K (note: timing discrepancies compared to Figure 5 occur due to testing on different servers). When K < 20, metrics all improve as K increases. However, when K > 20, recall continues marginal improvement and recision and F1-score slightly decline and begin oscillating. This deterioration stems from excessive error pairs entering the fine-grained matching stage. To balance computational efficiency and performance, we select K = 20 as the optimal parameter.

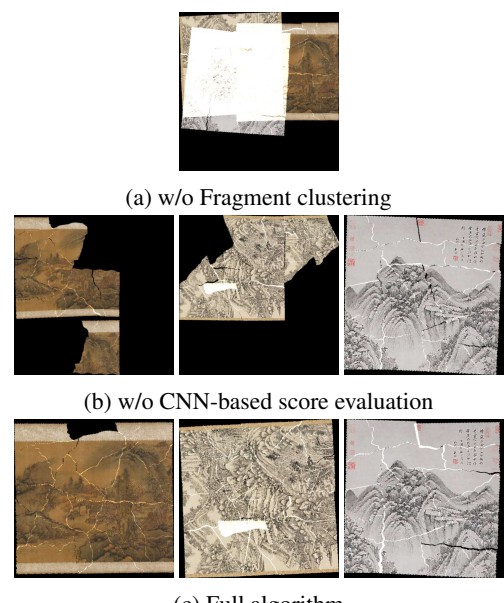

(a) w/o Fragment clustering

(b) w/o CNN-based score evaluation

(c) Full algorithm

Figure 13: As shown in the figure, when the fragment clustering module is removed, the system fails to correctly distinguish fragments belonging to different images. Similarly, when the CNN-based score evaluation module is removed, the system's performance noticeably declines.

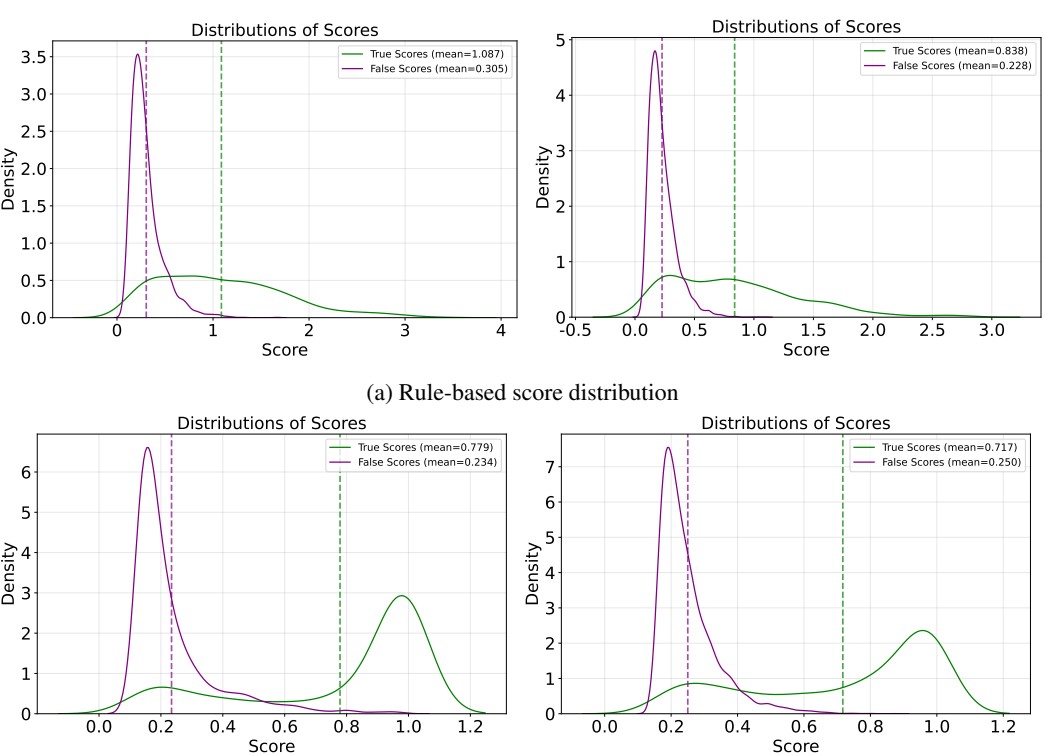

(a) Rule-based score distribution

(b) CNN-based score distribution

Figure 14: The left side of the figure shows the score distribution for the art_2192 dataset, while the right side shows the score distribution for the pex_2000 dataset. In each subplot, the horizontal axis represents the scores of fragment pairs, and the vertical axis represents the probability density.

Table 4: The global assembly precision and recall of two real-world cases. The first two columns represent the metrics for three real-world cases, while the last column represent the metrics for the synthetic dataset(art_2192).

| Cases | case1 | case2 | art_2192 |
|---|---|---|---|
| Prec(%) | 97.72 | 96.7 | 99.15 |
| Rec(%) | 98.2 | 97.73 | 97.59 |

Table 5: Results of the ablation studies.

| Module type | FM | | FC | GA | |
|---|---|---|---|---|---|
| | Prec(%) | Rec(%) | ARI(%) | Prec(%) | Rec(%) |
| w/o Fragment clustering | 76.71 | 79.01 | 53.44 | 97.94 | 97.31 |
| w/o fine-grained matching | 20.77 | **98.49** | 91.16 | N/A | N/A |
| w/o CNN-based score evaluation | 65.17 | 78.46 | 91.78 | 96.43 | 93.54 |
| Full algorithm | **77.07** | 78.40 | **92.29** | **99.15** | **97.59** |

We adopt the fragment feature extraction method from PairingNet, but to demonstrate that this method is more effective than directly applying pre-trained models, we compare our feature extraction with pre-trained models (ResNet50, ViT-B, ViT-L) on art_2192(3 images). The results are shown in Table 6. Our method outperforms these alternatives in fine-grained matching F1-scores.

Because manuscript restoration tasks prioritize detailed features near fragment contours, while other pre-trained models focus more on extracting semantic information from images, pre-trained feature extractors perform poorly.

We test our algorithm on subsets of art_2192 and pex_2000 with similar content. Specifically, we collect ShuiMo paintings from the art_2192 test set and portraits from the pex_2000 test set to form two new test sets, and evaluate the model on each. The global metrics obtained are shown in Table 7.

Both subsets contain images with high content similarity. The Global metrics demonstrate consistent performance regardless of content overlap, since the model leverages both contour and texture features for assembly. Even when two fragments exhibit nearly identical textures, the distinct contour features enable correct matching.

Our model is designed to be insensitive to fragment orientation and patch size. During training and testing, all fragments undergo random rotation from 0° to 360°. For patch size, since we used the feature extraction method of PairingNet which had conducted relevant studies and proved that 7x7 was the best patch size setting and insensitive to other patch sizes, we did not conduct repeated experiments and directly used the settings of PairingNet.

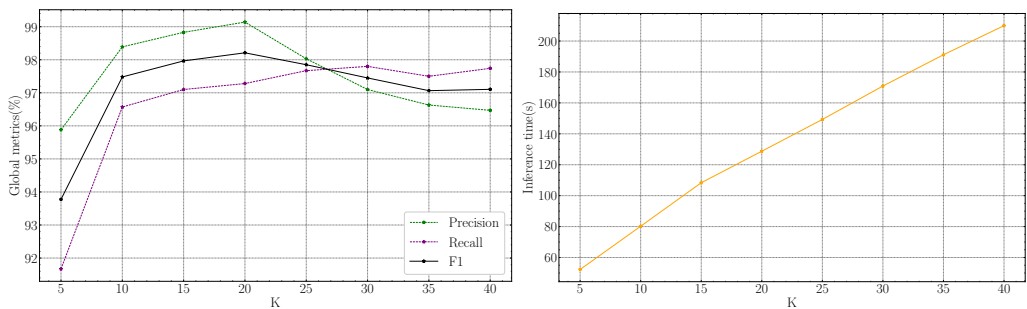

Figure 15: The changes in model performance(left) and time consumption(right) as K varies. The horizontal axis represents the values of K, and the vertical axis represents the Global Assembly metrics(left) and inference time(right).

Table 6: Fine-grained matching F1-scores of different feature extraction methods

| Model | ResNet50 | ViT-B | ViT-L | ResGAN(Ours) |
|-------|----------|-------|-------|--------------|
| F1-score(%) | 34.5 | 38.8 | 44.5 | **77.8** |

Table 7: Global metrics on the test set subsets of content-similar source images and the full test set.

| Dataset | art_2192 (ShuiMo) | art_2192 (all) | pex_2000 (portrait) | pex_2000 (all) |
|---------|-------------------|----------------|---------------------|----------------|
| Precision(%) | 99.23 | 99.15 | 98.37 | 98.2 |
| Recall(%) | 97.02 | 97.59 | 98.18 | 96.78 |

## J  THE USE OF LARGE LANGUAGE MODELS (LLMS)

We solely utilize LLMs to check for grammatical errors in the paper and to improve the fluency of its sentences. All other aspects of the work do not involve the use of LLMs.

