# OpenReview forum: "ShreddingNet: Coarse-to-Fine Restoration for Multi-Source Shredded Manuscripts"
_ICLR.cc/2026/Conference — ICLR 2026 Conference Withdrawn Submission_

### Official Review · Reviewer_5Hkc · 2025-10-28

**Soundness:** 2
**Presentation:** 2
**Contribution:** 3
**Rating:** 4
**Confidence:** 3

**Summary:**

The paper presents ShreddingNet, a coarse-to-fine pipeline for restoring multi-source shredded manuscripts. A coarse stage extracts texture/contour features per fragment, fuses them via cross-attention, and keeps Top-K candidate neighbors; label-propagation clusters fragments by source, assuming cross-source false matches are rare. A fine stage performs cross-attention at contour level and uses a lightweight CNN scorer plus RANSAC to validate matches, followed by MST-style global assembly. On large synthetic datasets (including degraded cases), the method reports strong precision/recall and claims near-linear scaling by constraining candidate pairs. The contribution is a clean integration of dual-modality features, graph clustering, and attentive matching tailored to multi-source reconstruction.

**Strengths:**

Pros:
(1) well-motivated task (multi-source) with a clear, modular design;
(2) solid headline gains over classic baselines;
(3) ablations indicate both clustering and CNN scoring matter;
(4) pragmatic assembly strategy.

**Weaknesses:**

Cons:
(1) external validity,evidence is largely synthetic; real-world variability (aging, illumination, non-rigid tears) is under-tested;
(2) “linear time” is argued via candidate scaling, not end-to-end runtime/memory curves across fragment counts;
(3) the rarity of cross-source false matches needs quantitative stress-tests;
 (4) comparative breadth, newer transformer/diffusion assemblers and significance testing are thin;
(5) presentation, more schematics/figures (failure modes, clustering dynamics, assembly flow) would substantially improve clarity and trust.

**Questions:**

see above

---

### Official Review · Reviewer_na2X · 2025-10-30

**Soundness:** 2
**Presentation:** 4
**Contribution:** 2
**Rating:** 4
**Confidence:** 4

**Summary:**

The paper introduces ShreddingNet, a two-stage, coarse-to-fine pipeline for multi-source fragment restoration, where fragments may originate from several different images. The method first performs a coarse matching using contour-texture fused graph features and cross-attention ResGCNs to find candidate fragment pairs efficiently, then clusters fragments by source through label propagation. Then, a fine-grained matching stage with stacked cross-attention and a CNN-based scorer refines pair alignments and reconstructs each source image via maximum-spanning-tree assembly. This design reduces complexity from quadratic to linear in fragment count and requires no prior knowledge of outlier ratios. On large synthetic datasets based on art_2192 and pex_2000 and degraded hard variants, ShreddingNet achieves SOTA performance. In addition, it runs substantially faster than prior SOTA methods such as PairingNet or JigsawNet.

**Strengths:**

1. The paper introduces a well-motivated pipeline for multi-source fragment restoration. The resulting linear-time complexity and ability to work without prior knowledge of the number of sources are good.
2. The writing is clear, well-structured, and technically precise, with definitions, equations, and figures that make the workflow easy to follow.

**Weaknesses:**

1. The coarse matching stage relies on the empirical observation that "wrong matches are intra-source." No detailed justification or overall error rate analysis is provided. This leaves the clustering reliability insufficiently analyzed.
2. From table 1, there is a noticeable gap between the fine-grained stage and the global assembly stage. This suggests that much of the high performance comes from graph-level correction rather than a robust pairwise matcher. This reliance may hide systematic fine-grained stage errors or over-smoothing.
3. The pipeline mainly combines known graph fragment encoding, cross-attention fusion, and CNN-based scoring. And due to the above two weaknesses, the pipeline's novelty is limited.

**Questions:**

Weakness 1, 2, and:
The assumption that "wrong matches are intra-source," if violated, will propagate errors throughout the pipeline. The paper would benefit from explicit error-correction mechanisms or uncertainty quantification to detect when this assumption fails. For hard-variant datasets, how much of the performance drop originates from the initial assumption's breakdown rather than the subsequent module? Moreover, how robust is the assumption under realistic conditions such as repeated patterns, high texture homogeneity, or large missing regions that obscure contour cues?

---

### Official Review · Reviewer_ZsZ5 · 2025-10-31

**Soundness:** 3
**Presentation:** 3
**Contribution:** 3
**Rating:** 6
**Confidence:** 4

**Summary:**

The paper proposes a system to recover the original image from a set of image segments, which can be used for artwork restoration. The pipeline is composed of two stages: the coarse matching stage first extracts image segments' global features by aggregating the edge image features and contour geometry features, and retrieves related image pairs. Then, a fine matching stage is applied to find pixel-level contour correspondences among the retrieved image pairs and filter the incorrect image pairs. The proposed method is evaluated on real-world datasets and shows its effectiveness in improving the image segment matching accuracy.

**Strengths:**

1. The pipeline is reasonable and well-designed. The two-stage coarse-to-fine matching strategy is effective in improving the image segment matching accuracy while improving the efficiency.
2. The overall writing is clear and easy to follow. The figures are well-organized and helpful in understanding the proposed method.

**Weaknesses:**

1. The paper only cites prior works on fragment feature extraction and matching works. However, the idea of coarse retrieval via global features and then finding pixel-level correspondences is widely used in other related fields, such as image retrieval and image matching.
The major difference is that this work focuses on finding correspondences on image contour segments rather than full images; however, the high-level idea is similar.
The reviewer thinks that the authors should reference these related fields.
Including but not limited to the following works:
From Coarse to Fine: Robust Hierarchical Localization at Large Scale;
SuperGlue: Learning Feature Matching with Graph Neural Networks;
Semi-dense feature matching with transformers and its applications in multiple-view geometry;
NetVLAD: CNN architecture for weakly supervised place recognition;

2. It's not clear how the proposed method handles the contour defects, such as broken or noisy contours. In real-world scenarios, the image segments may have imperfect contours due to damage or noise.
Since the fine matching stage relies on accurate contour correspondences for recovering the transformation between image segments, the contour defects may lead to incorrect matching results.
The reviewer suggests that the authors discuss how to handle these contour defects in the paper and provide some analysis on how these defects affect the matching performance.

**Questions:**

1. Contour binary maps are traded as images or some other data structure for feature extraction?

2. How many points are used for contour? It seems that using every contour pixel will lead to a significant number. Line 191 mentions that contour points exceeding L are truncated. Why use truncation instead of sampling? It seems that truncation may lose important shape information.

3. What are the benefits of incorporating image features? It seems that contour shape is more important for image segment matching. Can you ablate the image feature extraction module to see how it affects the performance?

---

### Official Review · Reviewer_Qhgm · 2025-11-01

**Soundness:** 3
**Presentation:** 2
**Contribution:** 2
**Rating:** 4
**Confidence:** 4

**Summary:**

The paper proposes ShreddingNet, a two-stage, coarse-to-fine pipeline for multi-source fragment reassembly. Stage-1 extracts contour–texture bimodal features, builds a fragment similarity matrix, keeps Top-K neighbors per fragment, and runs graph clustering (LPA) to separate sources; Stage-2 applies an inter-fragment cross-attention decoder to score candidate pairs, estimates local transforms (via RANSAC), and assembles globally with a maximum-spanning-tree strategy. The authors claim linear time complexity with respect to fragment count (by capping candidates per fragment) and robustness to stains, mold, and contour defects, reporting strong results on large synthetic datasets

**Strengths:**

• Well-structured, practical pipeline.
The decomposition—coarse Top-K pruning + source clustering, followed by fine pair scoring + MST assembly—is clean and addresses both scalability and accuracy in a realistic way. The architecture description is clear, and the use of cross-attention for feature fusion and inter-fragment interaction is appropriate to the fragment domain.

• Graph clustering to separate sources.
Leveraging the observation that most erroneous coarse matches remain intra-source enables effective LPA clustering without specifying the number of sources, a good fit for multi-source restoration.

•Efficiency story is plausible and backed by timing plots.
By constraining candidates to B×K, model invocations grow linearly with fragments; the timing regressions vs. image counts (Fig. 5) empirically show near-linear behavior on the two datasets.

•Comprehensive ablations and module-level reporting.
The paper reports module-wise precision/recall (CM/FM/SE/GA) and ablates clustering, the fine stage, and the score network; it also defines evaluation metrics (including ARI for clustering) carefully.

**Weaknesses:**

• Evidence for “linear time complexity” is empirical and conditional.
The claim hinges on a fixed K and the assumption that Top-K retains sufficient true neighbors as the number of fragments grows. The paper does not provide a formal argument for recall@K stability or bounds relating K, feature margins, and fragment counts. The linear fits in Fig. 5 are encouraging but do not constitute a complexity guarantee (they also depend on hardware and implementation constants). Clarify under what distributional conditions Top-K recall remains bounded away from zero and how worst-case adversarial configurations behave.

• External validity is limited by heavy reliance on synthetic data.
Training and testing are primarily synthetic; the real-world evaluation comprises two small case studies produced by printing/tearing/scanning a few images. It is unclear how the method fares on genuine museum scans (paper translucency, bleed-through, nonplanarity, illumination gradients). A larger real-world benchmark (≥50 images) or collaboration with a heritage institution would substantially strengthen the claims.

• Comparative coverage may be incomplete.
The baselines (rule-based, JigsawNet, PairingNet) are adapted for multi-source reassembly, and one relevant prior (LLMCO4MR) is excluded due to missing artifacts. The field also includes recent GNN/OT/transformer puzzle solvers and diffusion-based approaches that might be competitive after adaptation. Please expand baseline breadth or justify omissions with implementation and fairness details.

• Clustering reliability and failure cases are under-analyzed.
LPA is sensitive to graph edge noise and can produce inter-source bridges if coarse similarities are “sticky.” The paper argues errors are mostly intra-source, but does not quantify the rate and impact of bridge edges nor give sensitivity curves of ARI vs. K and vs. corruption level. Provide precision–recall for clustering labels (not just assembly edges) and analyze failure modes (e.g., content-similar sources).

• Score network calibration and domain shift.
A lightweight CNN + MLP scores similarity matrices; however, there is no discussion of calibration (ECE/Brier) or transfer to different materials/inks/languages. Since the score threshold is a key hyperparameter, present calibration plots and threshold sensitivity across domains (art vs. pex; normal vs. hard).

• Efficiency accounting across methods.
Timing includes all stages, but RANSAC cost can dominate in some pipelines; the paper argues its coarse-to-fine design reduces the number of RANSAC calls relative to PairingNet. Please provide per-stage time breakdowns and standard deviations, and clarify pre/post-processing overheads to ensure fair timing comparisons.

**Questions:**

• Coarse features and fusion (Sec. 3.2).
1) The paper limits contour points to L and asserts no truncation by “adjusting L,” but on pex 2000, L=2400—what is the distribution of contour point counts, and what is the runtime/memory sensitivity to L? Report histograms and ablate L vs. accuracy/time.
2) Cross-attention fusion averages two directional attentions. Why average rather than concatenate + projection or gated fusion? A small ablation here would help.

• Candidate selection & K.
The K-sweep in Appendix I is useful; consider plotting recall@K for true neighbors in Stage-1 separately from downstream F1, to quantify pure coarse recall. Also show fragment-count–normalized K (e.g., K as a fraction of B) to discuss scalability.

• Clustering (LPA).
Please report ARI vs. K on both datasets and under corruption (hard settings). Include bridge-edge rate (probability an inter-source edge survives after coarse matching).

• Fine-stage interaction (Sec. 3.3).
1) The cross-attention decoder depth is n=2. Provide a depth sweep (n = 1, 2, 4) and head count sweep; show trade-offs vs. runtime.
2) The valid mask on M removes off-contour entries—explain mask construction under contour defects (holes, extrusions).

• Score estimation.
Beyond the CNN vs. rule-based contrast, please provide ROC curves and AUC distributions by difficulty, plus calibration curves.

• RANSAC details
What is the inlier threshold and minimal set? Do you estimate affine, similarity, or homography? Provide robustness to outlier ratios and ablate RANSAC iterations.

• Global assembly (MST).
The MST weight mixes score and transform—how are these normalized and combined? Provide a weighting ablation and analyze cycles/contradictions when multiple high-score edges disagree geometrically.

• Metrics
You already include multiple seeds for Table 1 (great). Please add mean ± std for timing in Fig. 5, and run significance tests (paired) against the strongest baseline on overlapping test subsets.

• Dataset realism.
Synthetic stains/mold/contour defects are well documented, but they may not capture paper warp, nonplanarity, translucency. Consider simulating projective/distortion fields and illumination gradients, or at least discuss limitations.

• Real-world cases.
The two case studies suggest promise but are too small for statistical conclusions. Please release all scans, masks, and outputs to enable third-party replication; add a mini-benchmark (≥50 images) if feasible.

• Fairness in baseline adaptations.
You replace JigsawNet’s HLM assembly and choose arbitrary trees for PairingNet (no learned scores). Document these choices and report time/quality with at least one alternative fair setting per baseline to guard against adaptation bias.

• Reproducibility materials.
The paper claims code/models/data in the supplementary and on HF; ensure exact scripts (training, evaluation, data generation), random seeds, and failure cases are included.

---

### Note · Authors · 2025-11-12

I have read and agree with the venue's withdrawal policy on behalf of myself and my co-authors.